# MULTI-MODALITY ALONE IS NOT ENOUGH: GENERATING SCENE GRAPHS USING CROSS-RELATION-MODALITY TOKENS

## ABSTRACT

Recent years have seen a growing interest in Scene Graph Generation (SGG), a comprehensive visual scene understanding task that aims to predict the relationships between objects detected in a scene. One of its key challenges is the strong bias of the visual world around us toward a few frequently occurring relationships, leaving a long tail of under-represented classes. Although infusing additional modalities is one prominent way to improve SGG performance on under-represented classes, we argue that using additional modalities alone is not enough. We propose to inject entity relation information (Cross-Relation) and modality dependencies (Cross-Modality) into each embedding token of a transformer which we term primal fusion. The resulting *Cross-RElAtion-Modality (CREAM)* token acts as a strong inductive bias for the SGG framework. Our experimental results on the Visual Genome dataset demonstrate that our CREAM model outperforms state-of-the-art SGG models by around *20%* while being simpler and requiring substantially less computation. Additionally, to analyse the generalisability of the CREAM model we also evaluate it on the Open Images dataset. Finally, we examine the impact of the depth-map quality on SGG performance and empirically show the superiority of our model over the prior state of the art by better capturing the depth data, boosting the performance by a margin of around *25%*.

## 1 INTRODUCTION

Visual scene understanding has evolved in recent years from mere object detection and recognition tasks to more complex problems such as Visual Question Answering (VQA) (Antol et al., 2015) and Image Captioning (IC) (Hossain et al., 2019). One prominent tool for scene understanding is scene graph generation (SGG) (Lu et al., 2016): Given any two entities in a scene, the task of SGG is to detect any existing relationships between them. While standard SGG uses entity features from the RGB images to detect relations, to move towards the goal of generating scene graphs that adequately typify our visual world, we need additional clues to effectively capture under-represented classes in SGG. In this regard, researchers have taken multiple directions such as infusing complementary modalities (Zareian et al., 2020; Sharifzadeh et al., 2021) or conditioning using additional image context (Lu et al., 2021). Among SGG methods based on further modalities, Zareian et al. (2020) exploit external knowledge by using a late fusion mechanism in which the scene graphs generated from the RGB features are refined in multiple iterations per relation detection iteration using knowledge graphs, resulting in very high computational cost. On the other hand, SGG with additional depth data (Sharifzadeh et al., 2021) has used an early fusion mechanism. Although Sharifzadeh et al. (2021) requires less computation, it fails to effectively fuse the modalities and uses depth maps of limited quality (Fig. 7).

To effectively fuse different modalities, transformer models can be beneficial, with their usage expanding from text (Vaswani et al., 2017) to other modalities such as images (Dosovitskiy et al., 2021) and speech (Lin and Wang, 2020). Recently, transformers have also been used in SGG, mostly to capture dependencies across time in video-based SGG (Cong et al., 2021). In the case of still images, transformers were used to extract object dependencies (Dhingra et al., 2021) and context dependencies (Lu et al., 2021). Capturing known dependencies explicitly can boost the performance on under-represented classes (Lu et al., 2021). However, using transformers for multi-modal fusion

Figure 1: **Primal vs Early Fusion**. In the case of primal fusion the modalities and the subject-object features are fused explicitly to form the transformer input tokens. Contrastingly, for early fusion there is no interaction between the modalities and subject-object features prior to entering the transformer.

can be challenging because of the high computational cost and model complexity (Nagrani et al., 2021). The major reason for the increased expense in multi-modal transformers stems from the fusion strategies used. Although there is lot of research into multi-modal fusion, it has primarily focused on fusion strategies inside the transformers, which results in an increased sequence length thus increasing the model complexity. Moreover, explicitly modeling the known inductive bias is challenging for fusion strategies that start inside the transformer. The limitations of the existing methods raise the following open questions: *(Q1)* How can we effectively combine different modalities in SGG? *(Q2)* What is more important for improving SGG performance, better quality depth maps or a better fusion strategy? *(Q3)* Will having multiple modalities alone be enough to boost the coverage of under-represented classes in SGG?

To address the above questions, we propose a simple yet effective token generation strategy that we call *Cross-RElAtion-Modality (CREAM) tokens*, by strategically combining entity relations (between the features of the subject and the object entity) with modality dependencies (i.e. between RGB and the corresponding depth modality). As depicted in Fig. 1 we explicitly combine the modalities and subject-object features prior to entering the transformer unlike early fusion in which we rely on implicit fusion inside the transformer. We call our fusion strategy *primal fusion*. Surprisingly, our primal fused CREAM tokens are able to capture the inherent inductive bias in SGG using a single encoder-only transformer without using any complex cross-attention (Chen et al., 2021), fusion (Prakash et al., 2021), or encoder-decoder (Lu et al., 2021) architectures. In particular, we are using our system to study the effect of the depth-map quality in scene graph generation using an improved depth map, VG-Depth.v2, generated for the Visual Genome (VG) dataset (Krishna et al., 2017) using the monocular depth estimator of Yin et al. (2021) and compare to VG-Depth.v1 maps (Sharifzadeh et al., 2021) (Fig. 7). This study is crucial for two reasons: *(1)* to evaluate the relationship between architectural choice and depth-map quality in SGG performance and *(2)* for the efficacy of SGG in real-world visual scene understanding scenarios such as automated driving, where instantaneous 3D reconstruction is far-fetched but using monocular depth estimators to generate depth maps is feasible. We make our code publicly available at `https://anonymous.4open.science/r/CREAM_Model-113F`.

Precisely, we make the following contributions: *(C1)* We propose a novel token generation strategy (CREAM tokens) for transformer based multi-modal SGG. Our CREAM tokens explicitly force the multi-head-self-attention (MSA) component of the transformer to learn enriched representation by focusing on different subspace. Our proposal can significantly boost the performance while also reducing the computational cost. *(C2)* By primaly fusing CREAM tokens, we outperform state-of-the-art models on the mRecall metric despite not using any additional context. *(C3)* We conduct extensive depth-data analysis and ablation studies to show the significance of our proposed approach.

## 2 RELATED WORK

**SGG.** Scene graphs have been receiving increased attention from the research community due to their potential usability in assisting downstream visual reasoning tasks (Shi et al., 2019; Wang et al., 2019; Krishna et al., 2018). The scene graph generation (SGG) task was first introduced by Lu et al. (2016). The bias problem in SGG was first brought into focus by Zellers et al. (2018) and an unbiased evaluation metric, meanRecall (mRecall), was proposed by Chen et al. (2019a) and Tang et al. (2019). Various debiasing techniques were proposed subsequently such as using unbiased loss

functions including the class-balanced loss (Zareian et al., 2020), employing reinforcement learning (Lu et al., 2021), and disentangling biased representations from the unbiased (Misra et al., 2016; Cadene et al., 2019; Tang et al., 2020).

**Privileged Scene Graph Generators (PSGGs).** Recent progress in SGG pursues additional information sources to generate scene graphs beyond the standard subject-object features, labels, and bounding boxes. We call these models privileged SGG (PSGG) models. One such direction involves utilizing context information (Lu et al., 2021; Wang et al., 2019; Hung et al., 2020; Chen et al., 2019b). Lu et al. (2021) introduce a transformer-based SGG approach that propagates global context by making sequential prediction on the decoder by conditioning on the earlier decoded predictions. Through experiments, we show that CREAM model that uses depth as additional information instead of context, outperforms Lu et al. (2021). Another set of PSGGs use external knowledge graphs (KG) as additional information (Zareian et al., 2020; Chen et al., 2019a; Gu et al., 2019). KG-based SGG generally uses Graph Neural Networks for KG-SGG information propagation. Due to the complex architecture required for KG-based SGG, these models are highly computationally expensive. We show that our proposed approach outperforms KG-based SGG by a significant margin. Sharifzadeh et al. (2021) and Yang et al. (2018) have explored using depth maps as additional information source for SGG. However, the depth-maps used are of limited quality and the effective fusion of the depth modality with the RGB data has not been explored in detail. To address this, we analyse the impact of the depth-map quality on SGG performance and propose an effective way to combine the modalities and the entity features.

**Two stage and one stage SGGs.** The above works use two stage SGG pipeline in which the object proposals and relation detection are carried out at two separate stages using pretrained object detectors. On the other hand, one stage SGG (Li et al., 2022; Zou et al., 2021) generates the object proposals and entity relations in parallel. Though one stage SGG can produce sparse proposal set and faster models they are less generalisable. Thus, considering the generalisability and the availability of off-the-shelf object detectors we use the two stage SGG pipeline in our work.

## 3 PRIMAL FUSION: EARLY FUSION OF INDUCTIVE BIASES

To introduce primal fusion and to illustrate it using CREAM, let us briefly review the the core idea of the transformers exploited in our work and the significance of the proposed fusion strategy.

**Multi-Head-Self-Attention (MSA):** The core idea of our work is to effectively exploit the MSA component of the transformer (Lu et al., 2021). $\mathrm{MSA}(Q, K, V) = \mathrm{Concat}\left(\mathrm{SA}_1, \mathrm{SA}_2, \ldots, \mathrm{SA}_h\right) W^O$ with $\mathrm{SA}_i = \mathrm{softmax}\left(\frac{Q_i K_i^T}{\sqrt{d}}\right) V_i$, where SA refers to self-attention, $Q$, $K$, and $V$ refer to query, key, and value, and $W^O$ are the trainable parameters. The input token to the transformer is split into multiple heads to be attended to in parallel by the MSA.

The key to successfully capture the relationship between subject-object pairs in an SGG set-up is to effectively express the subject-object joint features that accounts for their intricate relationship. The MSA component of the transformer can jointly attend to diverse features by enabling each head to focus on different subspace with different semantic or syntactic meanings. Hence, we propose to use the MSA component to capture the inductive bias in the scene graph generation task. By enriching each token with both the modalities and subject-object features, we are exploiting the multi-head-self-attention component of the transformer effectively by feeding it with different subspace representations to learn from. As a result, the MSA with multiple heads can jointly attend to the diverse features which enables the transformer to learn richer representation.

**Multi-Model Fusion:** The question of where to fuse the modalities in a multi-modal setup is a pressing issue. Various fusion methods such as early, mid, late and bottleneck fusion were introduced (Nagrani et al., 2021). Early and mid fusion passes the token sequences of all the modalities as input to the transformer and the cross-modal information is exchanged implicitly starting from the initial or later layers in the model . In the case of late fusion, no cross-modal information is exchanged. The bottleneck fusion mechanism restrict the pairwise self-attention across specific units. Deviating from the above, our primal fusion explicitly fuse each embedding token with known dependencies, which acts as a strong inductive bias for SGG. In addition to performance gain the overall sequence length of the input tokens is reduced to single modality sequence while having enriched tokens.

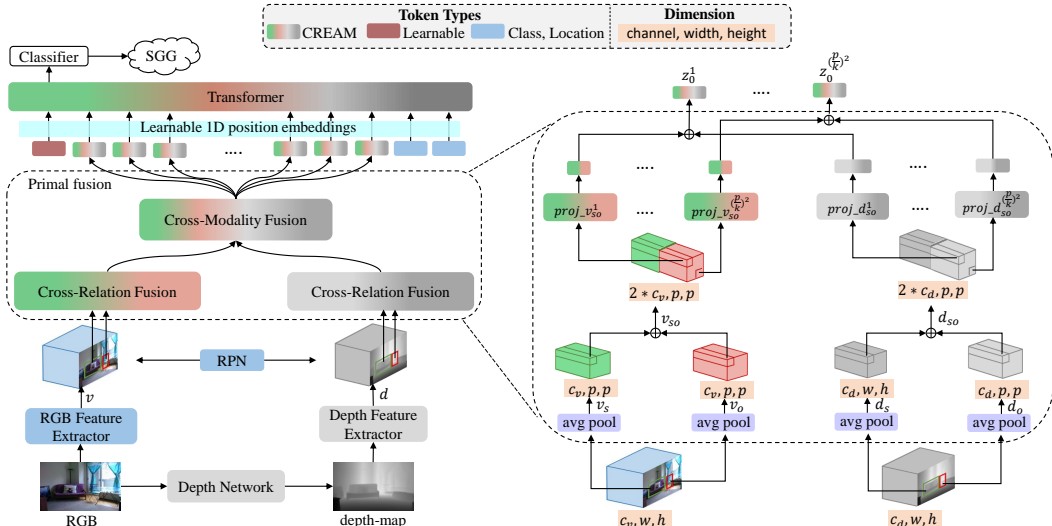

Figure 2: **The CREAM architecture**. An object detector, operating on each RGB image, first yields entity proposals and entity features $v$. Moreover, a depth map is estimated from the RGB input, which is also passed through the feature extractor to obtain depth features $d$. Then, for each subject-object pair, a sequence of Cross-Relation-Modality tokens are generated through Cross-Relation and Cross-Modality fusion of the subject-object features. The resultant tokens are passed through the transformer to yield a relation prediction.

## 3.1 THE CREAM ARCHITECTURE

Before describing the CREAM architecture let us briefly review the standard two-stage SGG pipeline that serves as the basis of our SGG approach. The first stage of a standard SGG pipeline (Lu et al., 2016) starts with image feature extraction followed by detection and classification of the entities in the image. The major components for relation detection between entities form part of the second stage. For any given pair of entities detected in an image, the second stage of a vanilla SGG model extracts the RGB features of the entities' bounding box regions (location), and projects these RGB features along with their location and class information. The final layers of the second stage then contain a multi-layer perceptron followed by a relation classification head.

Our CREAM architecture also comes under the two stage SGG pipeline. Fig. 2 shows the overall architecture of our proposed Cross-Relation-Modality transformer. For a given image $I$, we first estimate its depth map $D$ using a monocular depth estimator. We use Faster R-CNN (Ren et al., 2015) to generate object proposals on the RGB features extracted from $I$, restricted to the proposal region $v$. The same proposal information is used to extract the corresponding depth features $d$. The object proposal information constitutes the bounding boxes, also called the location information $l$, and the class probability distribution $c$. Followed by feature extraction, the input tokens for the transformer encoder are generated such that each token sequence contains information of a single subject-object pair. Each token contains primal fused information from a part of the subject-object pair from both RGB and depth modalities. The resultant tokens are passed through the transformer encoder-only model to predict the relations for the subject-object pair.

## 3.2 GENERATING CREAM TOKENS

Next, we describe the CREAM token generation strategy for a given pair of subject proposal $s$ and object proposal $o$. Let $c_v, w, h$ and $c_d, w, h$ be the channel, width, and height dimensions for $v$ and $d$, respectively. Further, let $v_s$ and $v_o$ be the RGB features of the subject and object region proposals, and let $d_s$ and $d_o$ be the corresponding depth features of the subject and object region proposals. Our primal fusion constitutes two stage fusion. In the first stage we average pool $v_s$, $v_o$, $d_s$, and $d_o$ along $h$ and $w$ to summarize the average presence of the features that are crucial for relation prediction. The resultant pooled features have height and width $p$ which we fuse channel-wise for each modality. Thus the first stage fusion comprise of $v_{so} = \text{fusion}_1(v_s, v_o)$ and $d_{so} = \text{fusion}_1(d_s, d_o)$, where $\text{fusion}_1(\cdot)$ denotes the pooled features that are combined along the channel dimension. Then we split

$v_{so}$ channel-wise into sequential patches such that if $c_v$, $p$, $p$, and $k$ are the channel, width, height, and patch size of $v_{so}$, we obtain $(p/k)^2$ patches for $v_{so}$. Similar to $v_{so}$, the fused depth features $d_{so}$ are also split into sequential patches such that we obtain $(p/k)^2$ patches for $d_{so}$ with channel size $c_d$. Each individual patch comprise of the summary of object features in a specific region, now by combining the subject and object patches together, we are making it explicit to the transformer that there are some feature dependencies between the subject and object patches that need to be exploited for the relation prediction. Hence, the first stage fusion corresponds to the Cross-Relation part of the token generation.

Note that this differs from Lu et al. (2021), who use transformers to capture relations between entities and treat each entity the same way as a text token in natural language processing. For an SGG setup, this assumption is limited because for effective relation detection we should capture dependencies between localised parts within and between entities rather than considering the entity as a whole. While splitting a word would make it meaningless, splitting an object would aid capturing its part compositions. Our Cross-Relation component overcomes this limitation by capturing the feature dependencies more effectively by facilitating the model to attend to crucial dependent features easily when passed through the transformer to predict the relation.

Followed by the Cross-Relation component creation, the second stage fusion starts where we linearly project the patches of both the modalities such that the depth modality has higher projected dimensionality than the RGB data, owing to the need to represent 3D structure: $proj\_v_{so}^i = f(v_{so}, p^v)$, $proj\_d_{so}^i = f(d_{so}, p^d)$, where $f(\cdot)$ denotes the linear projection, $p^v$ and $p^d$ the projection dimensionality of $v_{so}$ and $d_{so}$, and $i$ denotes the patch number ranging from 1 to $(p/k)^2$. Note that the sensitivity of our model to the exact proportion of depth and RGB dimensions in CREAM tokens is negligible in practice (Fig.5). After linear projection of all the patches, the corresponding patches of both the modalities are fused together. This reduces the total number of tokens from $((p/k)^2 * 2)$ to $(p/k)^2$. The second stage of fusion combines Cross-Relation patches of both the modalities, resulting in Cross-Relation-Modality token generation: $z_0^i = \text{fusion}_2(v_{so}^i, d_{so}^i)$, where $fusion_2(\cdot)$ denotes the modality specific projection $f(\cdot)$ followed by Cross-Modality fusion of dependant features. The modality dependency is based on the fact that every feature patch in the RGB modality has a corresponding feature patch in the depth modality. Thus the $fusion_2$ reduces the length of the token sequence while ensuring that dependent modality information is closely-knit to be effectively exploited by the Multi-head Self Attention (MSA) mechanism.

Additionally, the class and the location information obtained from the region proposal network for the subject-object pair are combined and linearly projected to replicate the Cross-Relation-Modality token dimension $p^v + p^d$. This results in two additional tokens: $z^{(p/k)^2+1} = h(f(l_{so}, p^v + p^d))$ and $z^{(p/k)^2+2} = h(f(c_{so}, p^v + p^d))$, where $f(\cdot)$ is the linear projection and $h(\cdot)$ is the non-linear function and $l_{so}$, $c_{so}$ denotes the location and class information of the subject-object pair. As we will see in Sec. 4, by using this Cross-Relation-Modality token generation strategy, our encoder-only model outperforms prior state-of-the-art models despite not using any cross-attention, complex fusion mechanism, or global context propagation.

## 3.3 TRANSFORMER ENCODER

We use a standard vision transformer encoder architecture (Dosovitskiy et al., 2021). We prepend the CREAM tokens with an additional learnable token $z_0^{class}$ and learnable 1D positional embedding $z^{pos}$ is added to each token to preserve the positional information. The resultant sequence is passed as input to the encoder. The transformer consists of $L$ encoder layers, each with MSA and MLP blocks. Each encoder layer also contains LayerNorm (LN) before every block and residual connections after every block:

$$z_0 = \left[z_0^{class}; z_0^1; \cdots; z_0^{(p/k)^2}; z_0^{(p/k)^2+1}; z_0^{(p/k)^2+2}\right] + z^{pos}, z^{pos} \in \mathbb{R}^{((p/k)^2+3)\times(p^v+p^d)} \quad (1)$$

$$z'_\ell = \text{MSA}(\text{LN}(z_{\ell-1})) + z_{\ell-1}, \quad \ell = 1 \ldots L \quad (2)$$

$$z_\ell = \text{MLP}(\text{LN}(z'_\ell)) + z'_\ell, \quad \ell = 1 \ldots L \quad (3)$$

$$y = \text{LN}(z_L^0). \quad (4)$$

Finally, $y$ is linearly projected to total number of predicate relations and forms our relation classification head.

Table 1: **mRecall results on the Visual Genome (VG) dataset** (the higher, the better). Context models have additional privilege to mitigate errors propagated from classification (SGCLS) and detection (SGDET) unlike the knowledge graph (KG) and depth-privileged models. Hence, our major baselines for SGCLS and SGDET tasks are based on KG and depth. The best results overall on the SGCLS and SGDET tasks are denoted by the superscript *.

| Model | Privilege | PRDCLS mRecall@ | | | SGCLS mRecall@ | | | SGDET mRecall@ | | |
|---|---|---|---|---|---|---|---|---|---|---|
| | | 20 | 50 | 100 | 20 | 50 | 100 | 20 | 50 | 100 |
| MOTIFS (Zellers et al., 2018) | context | 10.8 | 14.0 | 15.3 | 6.3 | 7.7 | 8.2 | 4.2 | 5.7 | 6.6 |
| FREQ (Tang et al., 2019) | context | 8.3 | 13.0 | 16.0 | 5.1 | 7.2 | 8.5 | 4.5 | 6.1 | 7.1 |
| RTN (Koner et al., 2020) | context | – | – | 20.3 | – | – | 12.6* | – | – | – |
| RU-Net (Lin et al., 2022) | context | – | – | 24.7 | – | – | 13.9 | – | – | 10.1* |
| VCTree (Tang et al., 2019) | context | 14.0 | 17.9 | 19.4 | 8.2* | 10.1* | 10.8 | 5.2* | 6.9* | 8.0 |
| KERN (Chen et al., 2019a) | KG | – | 17.7 | 19.2 | – | 9.4 | 10.0 | – | **6.4** | 7.3 |
| GB-NET (Zareian et al., 2020) | KG | – | 19.3 | 20.9 | – | 9.6 | 10.2 | – | 6.1 | **8.3** |
| Depth-VRD (Sharifzadeh et al., 2021) | depth | 16.4 | 20.7 | 22.7 | – | – | – | – | – | – |
| CREAM (ours) | depth | **20.3** | **25.8** | **28.1** | **8.0** | **9.7** | **10.5** | 4.0 | 5.2 | 6.5 |

**Training.** For RGB and depth feature extraction, we follow the procedure of our baseline models (Zellers et al., 2018; Sharifzadeh et al., 2021). We train our model using the Adam optimizer (Kingma and Ba, 2015) with a batch size of 4 images and a learning rate of $1e-3$. For our transformer model, we use a patch size of 2, pooled entity resolution of 8, leading to a total of 19 input tokens $\left(((p/k)^2+3) \text{ where } p=8, k=2\right)$ with embedding size of 576; we employ 6 attention heads and 6 encoder layers. The model is trained for 45 epochs. We use cross-entropy as the loss function $\mathcal{L}$. To compare our model against previous debiased SGG models, we use a modified cross-entropy loss that uses importance weighting for each class as proposed by Cui et al. (2019), which was also used in Zareian et al. (2020). Specifically, we use the following class balanced cross-entropy loss: $\mathcal{L}_{\mathbf{CB}}(\mathbf{p}, y) = \frac{1-\beta}{1-\beta^{n_y}}\mathcal{L}(\mathbf{p}, y)$, where $\mathbf{p}$ denotes the predicted relation probabilities, $y$ denotes the ground-truth relation, $n_y$ the number of samples in the ground-truth relation $y$, and $\beta$ denotes the hyperparameter. $\beta = 0$ would result in the standard cross-entropy loss, whereas increasing $\beta$ strenghtens the debiasing. We used the setting from Zareian et al. (2020) for our class-balanced loss.

## 4 EXPERIMENTAL EVALUATION

We use the SGG benchmark of the Visual Genome (VG) dataset (Krishna et al., 2017) for our evaluation and the split proposed by Xu et al. (2017). We conducted our experiments on 3 tasks: *(1)* predicate classification (PRDCLS), *(2)* scene graph classification (SGCLS), and *(3)* scene graph detection (SGDET). To ensure a fair comparison against leading baselines, we created two variants of our models: *(1)* CREAM and *(2)* CREAM ($\beta$-loss). While both the models are compared against leading PSGGs in general, the CREAM ($\beta$-loss) uses a balanced cross-entropy loss ($\beta$-loss) as a debasing strategy to yield a fairer comparison against leading baselines that also uses additional debiasing strategies. We compare against PSGGs with additional context, knowledge graphs and further modalities. We trained each of our models 5 times with different random seeds. The observed variation in performance across training was very small with a variance of 0.20.

**Results and analysis.** mRecall results without using any debiasing strategy are shown in Table 1. For a fair comparison, all the baselines that are chosen for this comparison also do not use any debiasing strategy. We notice that our approach, denoted as CREAM, improves over the prior state of the art by more than 20% on the predicate classification task. For the scene graph classification and detection tasks, our model outperforms both the depth and knowledge graph-based SGG approaches. It can also be noticed that the models with additional context information show slightly better performance on the SGCLS snd SGDET tasks. This is expected because image context can provide additional clues to the model as it is independent of the classification score and the region proposals from the pre-trained Faster R-CNN (Ren et al., 2015) for any given subject-object pair. On the other hand, the extracted depth features are based on the same region proposals from Faster-RCNN for a given subject-object pair, and hence the errors get propagated. One interesting future work direction to enhance performance further would be to bring in context into our CREAM tokens.

Table 2: **mRecall results for debiased models on Visual Genome (VG)** (the higher, the better). Context models have additional privilege to mitigate errors propagated from classification (SGCLS) and detection (SGDET) unlike knowledge graph (KG) and depth-privileged models. Hence, our major baseline for SGCLS and SGDET tasks is based on KG. The best results overall on the SGCLS and SGDET tasks are denoted by the superscript *.

| Model | Privilege | PRDCLS mRecall@ | | | SGCLS mRecall@ | | | SGDET mRecall@ | | |
|---|---|---|---|---|---|---|---|---|---|---|
| | | 20 | 50 | 100 | 20 | 50 | 100 | 20 | 50 | 100 |
| MOTIFS+TDE:GATE (Tang et al., 2020) | context | 18.5 | 24.9 | 28.3 | 11.1 | 13.9 | 15.2 | 6.6 | 8.5 | 9.9 |
| VTransE+TDE:GATE (Tang et al., 2020) | context | 18.9 | 25.3 | 28.4 | 9.8 | 13.1 | 14.7 | 6.3 | 8.5 | 10.2 |
| VCTree+TDE:SUM (Tang et al., 2020) | context | 18.4 | 25.4 | 28.7 | 8.9 | 12.2 | 14.0 | 6.9 | 9.3 | 11.1 |
| Seq2Seq: RL-loss (Lu et al., 2021) | context | 21.3 | 26.1 | 30.5 | 11.9* | 14.7* | 16.2 | 7.5* | 9.6 | 12.1 |
| BGNN: multitask loss (Li et al., 2021) | context, resampling | – | 30.4 | 32.9 | – | 14.3 | 16.5* | – | 10.7* | 12.6* |
| GB-NET:$\beta$-loss (Zareian et al., 2020) | KG | – | 22.1 | 24.0 | – | 12.7 | 13.4 | – | 7.1 | 8.5 |
| CREAM:$\beta$-loss (ours) | depth | **25.1** | **31.6** | **34.8** | **10.4** | **12.8** | **14.0** | **5.8** | **7.9** | **9.1** |

Table 3: **mRecall results for PRDCLS task on Open Images v4 and v6** (the higher, the better). Baseline results from official code are denoted by the superscript †.

| Model | Privilege | PRDCLS (v4) mRecall@ | | | PRDCLS (v6) mRecall@ | | |
|---|---|---|---|---|---|---|---|
| | | 20 | 50 | 100 | 20 | 50 | 100 |
| MOTIFS[†] (Zellers et al., 2018) | context | 83.6 | 84.2 | 84.4 | 70.3 | 70.5 | 70.5 |
| Depth-VRD[†] (Sharifzadeh et al., 2021) | depth | 80.2 | 80.4 | 80.4 | 69.8 | 70.0 | 70.0 |
| CREAM(ours) | depth | **92.2** | **92.4** | **92.4** | **76.0** | **76.2** | **76.3** |

To compare against models that additionally use unbiasing strategies, we have created a variant of our model as shown in Table 2. While there are various debiasing strategies (Zareian et al., 2020; Tang et al., 2020; Lu et al., 2021), our motivation is to show the effectiveness of our proposed architecture in effectively utilising relation and modality dependencies. Hence, we chose a simple debiasing technique that uses a class-balanced cross entropy loss ($\beta-$loss). More complex debiasing strategies may result in further improvement in performance; we leave that for a potential future study. As discussed earlier, our biggest improvements are expected for the predicate classification task. Table 2 confirms once more that also our debiased model outperforms prior state-of-the-art debiased models. It should also be noticed that our model outperforms BGNN (Li et al., 2021) that additionally does bi-level data re-sampling in addition to debiasing and using context.

Fig. 3 depicts the class-level recall performance for our CREAM model in comparison to the Depth-VRD (Sharifzadeh et al., 2021). Both the models are trained using the VG-Depth.v2 dataset (see below). The yellow line denotes the long-tail of under-represented classes in the VG dataset (Krishna et al., 2017). It can be noticed that when we get closer to the tail, our CREAM model has higher recall values per class than Depth-VRD, showing how our CREAM tokens more effectively exploit the modality dependencies compared to prior depth-based multi-modal SGG approaches. The trend in Fig. 3 is also confirmed by the qualitative samples in Fig. 4. While Depth-VRD captures widely occurring relations such as *'wearing'* and *'on'*, it fails to capture rare spatial relations in the data such as *'sitting on'*, *'covered in'*, and the co-occurring triplet *'Table and chair'*. Our CREAM model's success at capturing both the rare spatial and co-occurring relations (highlighted in green) shows the effectiveness of our Cross-Relation and Cross-Modality combined CREAM tokens.

To investigate on the generalisability of our CREAM model we further evaluate it on Open Images v4 and v6 dataset (Kuznetsova et al., 2020). We use the same data split and backbone used by Li et al. (2021) and compare the mRecall performance on the PRDCLS task. As depicted in Table 3 CREAM model consistently outperforms the baseline models once again proving its effectiveness.

**Depth-map analysis.** We analysed the significance of the depth-map quality on the SGG performance and the importance of careful architectural design to make use of the depth map efficiently. As depicted in Table 4a, the performance improvement for our model on VG-Depth.v2 generated using the monocular depth estimator of Yin et al. (2021) over VG-Depth.v1 (Sharifzadeh et al., 2021) is more than 20% for all the mRecall thresholds. To the contrary, the model of Sharifzadeh et al. (2021) shows no noticeable improvement on the high quality VG-Depth.v2 dataset. This reveals that the improved performance of our model is not just due to the high-quality depth maps (VG-Depth.v2) but due to our CREAM architecture, which allows to take advantage of the improved

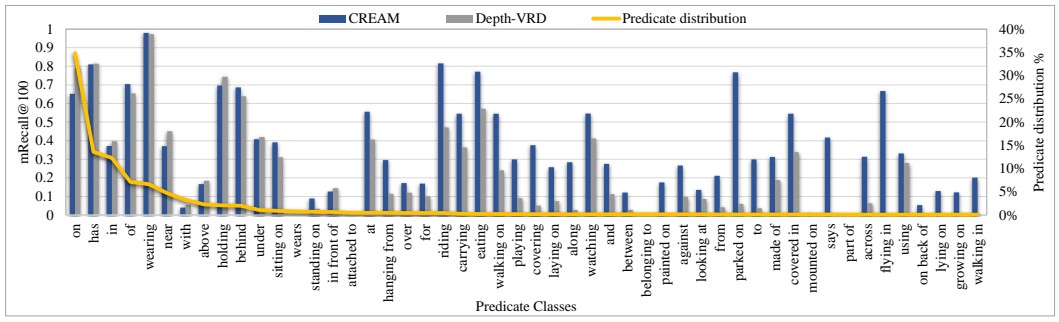

Figure 3: **Class-level recall@100 for PRDCLS**. The yellow line depicts the data distribution per predicate class. Our CREAM model, denoted using blue bars, has a higher recall than Depth-VRD (Sharifzadeh et al., 2021) for under-represented classes.

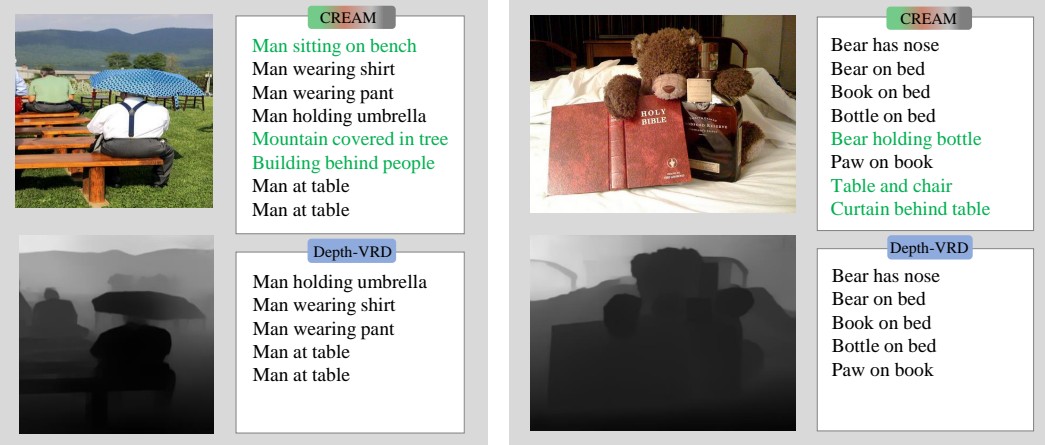

Figure 4: **Qualitative results for PRDCLS**. The triplets correctly predicted by CREAM while missed by Depth-VRD (Sharifzadeh et al., 2021) are highlighted in green. The CREAM model is successful at predicting both spatially-aware scenarios such as *'sitting on'*, *'covered in'* (left) and relation-aware scenarios *'and'* (right) missed by Depth-VRD.

depth maps in an effective manner. Also, our model's results on VG-Depth.v1 reveal the quality limitations of VG-Depth.v1, as despite the improved modeling power of our CREAM architecture, there is no significant mRecall improvement on VG-Depth.v1.

**Computational cost.** We compare the computational cost of our model against leading baselines in Table 4b. For fair comparison, all the models are timed on single NVIDIA A100 GPU with 40 gigabytes of memory. The most crucial aspect is the inference time of the model when it is deployed in real-time scenarios. Our model is 20 times faster than GB-NET (Zareian et al., 2020) while being significantly better at capturing under-represented classes. It also has around 35% and 3% fewer parameters than GB-NET and Depth-VRD, respectively.

Table 4: **Impact of depth-map quality and computational cost** of CREAM vs. the state of the art.

| Model | mRecall@ | VG-Depth.v1 | VG-Depth.v2 | Improvement |
|---|---|---|---|---|
| CREAM (ours) | 20 | 16.7 | 20.3 | **22%** |
| | 50 | 20.9 | 25.8 | **23%** |
| | 100 | 22.7 | 28.1 | **24%** |
| Depth-VRD | 20 | 16.4 | 16.0 | -2% |
| | 50 | 20.7 | 20.9 | 1% |
| | 100 | 22.7 | 23.7 | 4% |

(a) mRecall performance improvement on Visual Genome (VG) for PREDCLS (higher, the better).

| Model | Training Time (min/epoch) | Inference Time (ms/image) | #Parameters (million) |
|---|---|---|---|
| GB-NET | 180 | 500 | 444 |
| Depth-VRD | 28 | **25** | 298 |
| CREAM (ours) | **26** | 25 | **288** |

(b) Computational cost in comparison to the state of the art.

Table 5: **Ablation study**. mRecall results on Visual Genome (VG) for $\beta$-loss model variants (the higher, the better). The token design of the model variants is considerd $v$: visual, $d$: depth, $s$: subject, and $o$: object information; '-' denotes separate tokens.

| Model Variants | Model Description | PRDCLS mRecall@ | | |
| --- | --- | --- | --- | --- |
| | | 20 | 50 | 100 |
| *vso* | Cross-Relation (no depth) | 15.7 | 19.43 | 21.0 |
| *vso-dso* | Cross-Relation | 21.2 | 25.5 | 27.5 |
| *vds-vdo* | Cross-Modality | 17.1 | 21.4 | 22.0 |
| *vs-vo-ds-do* | no cross tokens | 16.7 | 20.0 | 21.4 |
| CREAM (*vdso*) | Cross-Relation-Modality | **25.1** | **31.6** | **34.8** |

**Ablation study.** To evaluate the importance of our CREAM token design, we conducted an extensive ablation study as shown in Table 5. Here, *vso* represents a model variant without using the depth modality such that every token in the sequence contains only RGB-based information about the subject-object pair. *vso-dso* denotes the model that uses both depth and RGB modalities but each token in the sequence contains either depth or RGB subject-object pair information and not both the modalities. *vds-vdo* denotes a model in which each token contains information about both depth and RGB modalities but either only subject or only object information. Finally, *vs-vo-ds-do* denotes the model in which the modalities and relations are all split into separate tokens. As Table 5 shows, the removal of depth features results in the highest performance drop. Moreover, Cross-Relation seems to be the most crucial fusion combination that outperforms other model variants. The effectiveness of Cross-Relation fusion strengthens our claim that for effective relation detection it is crucial to capture the dependencies between localised parts within and between entities. We can also notice that the standard transformer token variant that does early fusion (*vs-vo-ds-do*) improves only slightly over the no-depth model *vso*. Moreover, we find that Cross-Modality fusion (*vds-vdo*) results in an improvement over no-cross tokens model. Overall, our CREAM model that exploits Cross-Relation and Cross-Modality fusion significantly outperforms all other model variants.

Now we are ready to answer the open questions raised in the Introduction 1:

*Q1. Effective way to combine different modalities in SGG:* Through strong quantitative and qualitative results of our CREAM model discussed above, we prove that explicitly capturing the known inductive biases is one way to effectively combine different modalities in SGG.

*Q2. Better quality depth maps vs better fusion strategy:* From the depth data analysis in Table 4a we can conclude that it is crucial to have both better depth maps and better fusion strategy to improve the SGG performance.

*Q3. Boosting the coverage of under-represented classes in SGG using multi-modality alone:* Our ablation study results in Table 5 proves that multi-modality alone is not enough, effectively combining the Cross-Relation and Cross-Modality information is crucial at boosting SGGs performance on under-represented classes.

## 5 CONCLUSION

In this paper, we show that simply relying on multiple modalities is not enough to cope with under-represented classes in SGG. Careful attention must be paid to how the modalities and features are processed. We propose an effective way of moving towards multi-modality in SGG by introducing CREAM tokens, which enables the model to learn richer representation by explicitly capturing the inductive bias in an SGG set-up. We outperform leading SGG baselines by feeding CREAM tokens to the transformer using our primal fusion technique. Through an extensive ablation study and depth data analysis, we show the significance of our proposed approach. Though our model shows significant performance improvements on the PREDCLS task, to further mitigate the classification and detection errors in SGCLS and SGDET tasks we will have to bring in additional image context. One interesting avenue for future work would be to enrich CREAM tokens with context. Another promising research direction would be to expand our CREAM architecture to an end-to-end relation system that unifies Multi-Modality and Cross-Modality correlations.

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

# A    APPENDIX

This section contains additional results and analysis that were excluded from the main document due to space constraints. At first we carry out a sensitivity analysis of the modality proportion in the CREAM token. Then we discuss the mRecall results across model runs to show the stability of our model. We conclude by providing the Recall results and additional qualitative samples.

## A.1    SENSITIVITY ANALYSIS

We conducted a modality proportion sensitivity analysis as depicted in Fig. 5 by varying the number of projection dimensions of RGB $v_{\text{dim}}$ and depth $d_{\text{dim}}$, which in total sum to a dimensionality of 576. Starting with $v_{\text{dim}} = 64$ and $d_{\text{dim}} = 512$, the relative proportion is varied across runs by increasing $v_{\text{dim}}$ by 64 and decreasing $d_{\text{dim}}$ by 64 incrementally. The seed of all the model variants is fixed to 0 and the resulting mRecall@20, 50, and 100 are summed to get the best overall results. In Fig. 5, we observe that increasing $v_{\text{dim}}$ and reducing $d_{\text{dim}}$ from left to right shows an overall decreasing trend. This trend could be attributed to the fact that the depth modality requires a higher number of projected dimensions to represent the high-quality 3D information. Note that this hypothesis is consistent with the increased file size of the depth maps in comparison to the RGB data. Moreover, we observe that our CREAM architecture seems not to be particularly sensitive to the modality proportion. We have used $v_{\text{dim}} = 64$ and $d_{\text{dim}} = 512$ in all our CREAM model variants.

## A.2    MODEL STABILITY

In the main document we have reported the average of five model runs obtained by using random seeds 0 to 4. Fig. 6 depicts the mRecall results obtained for each of those five model runs. Our proposed CREAM model is highly stable with mRecall@20, 50, and 100 on the test set ranging from 24.3 to 25.8, 31.2 to 32.2, and 34.2 to 35.3, respectively. The mRecall results thus have low variances of 0.27, 0.18, and 0.17, respectively.

## A.3    RECALL RESULTS

Though the major focus of our work is to improve the coverage of under-represented classes by using mRecall as the evaluation metric we additionally analysed our models performance on Recall metric. Similar to the observation of previous works Lu et al. (2021); Tang et al. (2020); Zareian et al. (2020) we also noticed that optimising for Recall results in degradation of mRecall and vice versa. While the optimal mRecall results were obtained at the learning rate of $1e-3$, optimal Recall results were obtained at the learning rate of $1e-4$. Table 6 shows that our Recall results obtained at a learning rate of $1e-4$ are also competitive with the state of the art for the PREDCLS task. At $1e-3$ learning rate we noticed around 1% drop in Recall results reported in Table 6.

## A.4    VG-DEPTH.V1 VS VG-DEPTH.V2

Figure 7 shows "noisy" depth-map samples reported in Sharifzadeh et al. (2021). Comparing the noisy depth maps from Sharifzadeh et al. (2021) (VG-Depth) from the second row to the corresponding high-quality depth maps (VG-Depth.v2) in the bottom row.

## A.5    QUALITATIVE SAMPLES

To show the effectiveness of our method qualitatively, Fig. 8 and 9 compare scene graphs generated by our method to those generated by the Depth-VRD Sharifzadeh et al. (2021) baseline model. For fairness, both the models use our improved VG-Depth.v2 depth maps. Though both the models uses the same high quality depth maps, our model can effectively capture under-represented classes (classes represented in less than 5% of samples in the dataset) such as 'over', 'sitting on', 'laying on', 'behind' (Fig. 8) and 'covering', 'standing on', 'walking in' (Fig. 9). An additional description can be found in the respective figure caption.

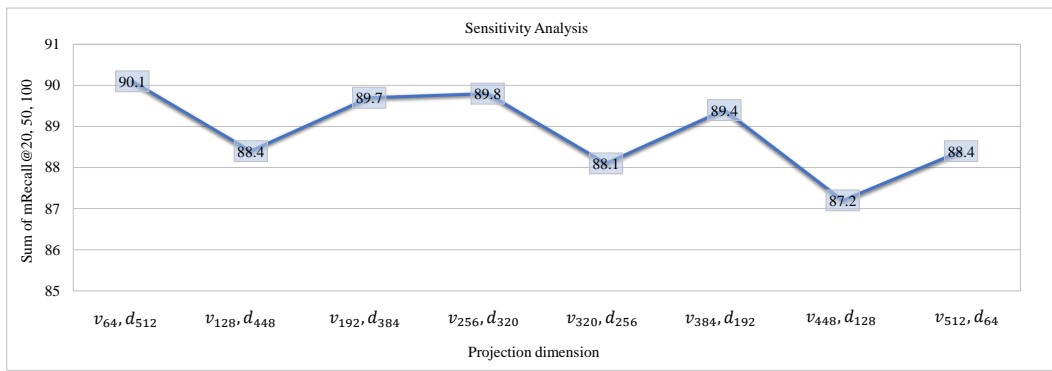

Figure 5: **Sensitivity analysis by varying the modality proportion within the CREAM tokens.** From left to right $v_{dim}$ is incremented by 64 while $d_{dim}$ is decremented by 64. We plot the sum of mRecall@20, 50, and 100.

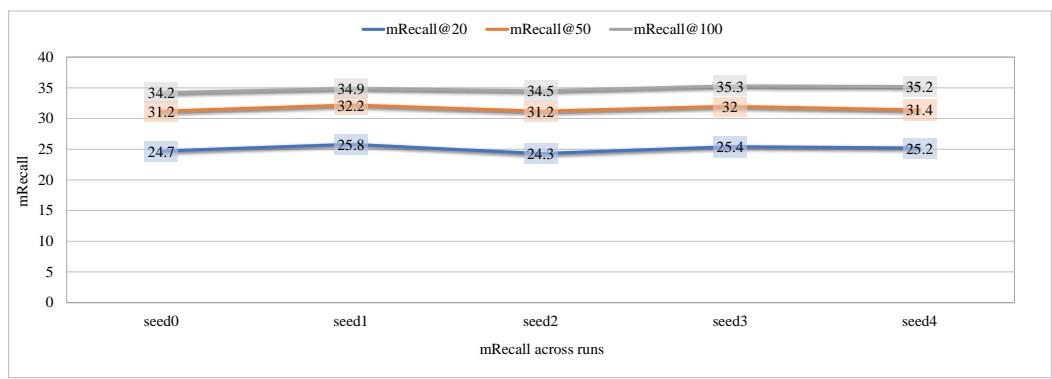

Figure 6: **CREAM($\beta$-loss): Variation across training runs.** mRecall@20, 50, and 100 on the test set is plotted for all the model variants obtained by varying the seed from 0 to 4.

Table 6: **Visual Genome (VG) recall results** (the higher, the better).

| | PRDCLS | | | SGCLS | | | SGDET | | |
|---|---|---|---|---|---|---|---|---|---|
| Recall@ | 20 | 50 | 100 | 20 | 50 | 100 | 20 | 50 | 100 |
| MOTIFS Zellers et al. (2018) | 58.5 | 65.2 | 67.1 | 32.9 | 35.8 | 36.5 | 21.4 | 27.2 | 30.3 |
| Frequency+Overlap Zellers et al. (2018) | 53.6 | 60.6 | 62.2 | 29.3 | 32.3 | 32.9 | 20.1 | 26.2 | 30.1 |
| Message Passing+ Zellers et al. (2018) | 52.7 | 59.3 | 61.3 | 31.7 | 34.6 | 35.4 | 14.6 | 20.7 | 24.5 |
| VCTree Tang et al. (2019) | **60.1** | 66.4 | 68.1 | **35.2** | **38.1** | **38.8** | **22.0** | 27.9 | 31.3 |
| GB-NET Zareian et al. (2020) | − | **66.6** | **68.2** | − | 38.0 | **38.8** | − | **26.4** | **30.0** |
| Depth-vrd Sharifzadeh et al. (2021) | 59.4 | 66.2 | 68.0 | − | − | − | − | − | − |
| CREAM | 59.3 | 66.3 | 68.1 | 31.3 | 34.5 | 35.3 | 17.0 | 21.9 | 25.0 |

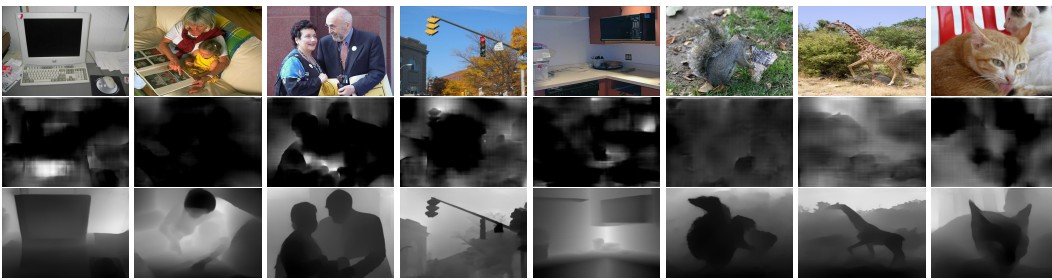

Figure 7: Failure cases reported by Sharifzadeh et al. (2021). The second row shows the noisy depth maps VG-Depth.v1 from Sharifzadeh et al. (2021). The bottom row represents the improved depth map VG-Depth.v2.

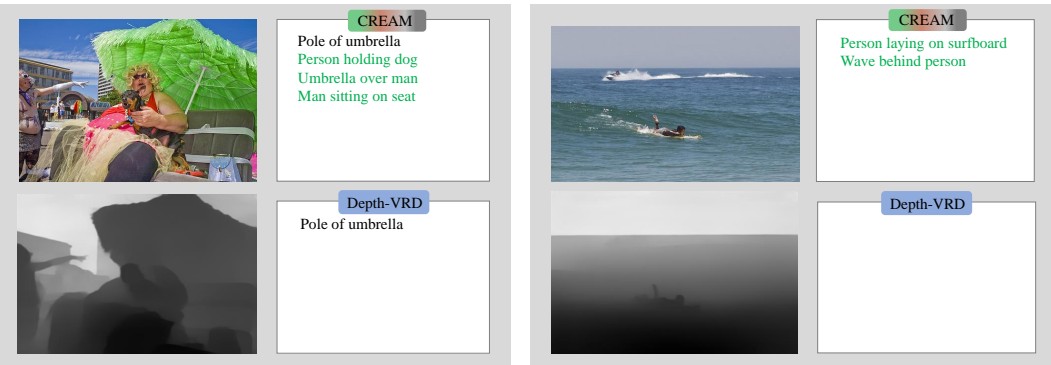

Figure 8: **Qualitative results for PRDCLS**. The triplets correctly predicted by CREAM while missed by Depth-VRD Sharifzadeh et al. (2021) are highlighted in green. The CREAM model successfully detects the under-represented classes 'over', 'sitting on' (left) as well as 'laying on', 'behind' (right) missed by the Depth-VRD model.

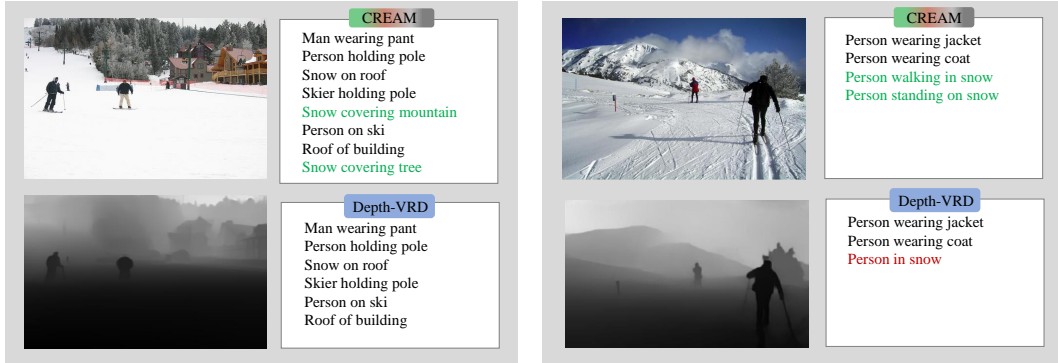

Figure 9: **Qualitative results for PRDCLS**. The triplets correctly predicted by CREAM while missed by Depth-VRD Sharifzadeh et al. (2021) are highlighted in green. Our proposed CREAM model successfully detects the under-represented class 'covering' (left) missed by the Depth-VRD model. The figures on the right show that our model can capture the descriptive under-represented classes 'walking in', 'standing on' (highlighted in green), while Depth-VRD could only relate the same pair using the highly-represented, less descriptive relation 'in' (highlighted in red).

