# OpenReview forum: "Multi-Modality Alone is Not Enough: Generating Scene Graphs using Cross-Relation-Modality Tokens"
_ICLR.cc/2023/Conference — Submitted to ICLR 2023_

### Official Review · Reviewer_JG3W · 2022-10-18

**Confidence:** 4
**Correctness:** 2
**Technical Novelty And Significance:** 2
**Empirical Novelty And Significance:** Not applicable
**Recommendation:** 3

**Clarity, Quality, Novelty And Reproducibility:**

The writing of the paper is clear. In my opinion, the paper missed a crucial point by ignoring the recent and relevant literature.

**Strength And Weaknesses:**

SGG is essentially a long-tailed and multi-task problem as it deals with two long-tailed distributions, namely entities and predicates. The evaluation metrics should not only emphasize on the head classes, but also on middle or tail classes. Thus, mean Recall should be more highlighted than recall. The paper follows the right directions from that perspective.

Although depth can provide complementary information, but given the combinatorial nature of SGG, the context shouldn't be ignored.

More importantly, the paper hasn't done the prior art survey on this very topic. Plenty of methods already exist that perform significantly better on SGG tasks as measured by the mean Recall metric. In fact, two popular methods that specifically emphasized the importance of mean Recall  are TDE (CVPR 20) and KERN (CVPR 19). Although these two papers were referred here but not compared in the main table. References such as PCPL (ACM MM 2020), DT2 (ICCV 2021), which notably improved on mean Recall metrics on Visual Genome, are missing.

The comparative performance as shown in Table 3 is for PredCls, where the set of all objects with their corresponding classes and bounding boxes are given. I'd rather like to look at such metrics for SGCls or SGDet.





**Summary Of The Paper:**

The core idea of the paper is to explore how to combine different modalities such as RGB and depth to improve scene graph generation tasks, specially to improve the performance on data-scare classes. The authors introduced the idea of combining subject-object relations with modality dependencies. To that end, they came up with a token generation strategy, termed as CREAM. They show the SGG task can utilize depth as another modality instead of utilizing context, or external knowledge graphs. Essentially the method is a two-stage one.
The authors claim that their method achieve state-of-the-art performance on Visual Genome for mean Recall metrics.

**Summary Of The Review:**

The paper hasn't done the prior art survey on this very topic. In my opinion, the paper missed a crucial point by ignoring the recent and relevant literature. The claim about outperforming existing methods maynot be a correct one.

---

### Official Review · Reviewer_XdXf · 2022-10-21

**Confidence:** 4
**Correctness:** 3
**Technical Novelty And Significance:** 3
**Empirical Novelty And Significance:** 2
**Recommendation:** 3

**Clarity, Quality, Novelty And Reproducibility:**

+ Clarity: The writing quality needs to be further improved. For example, the technical details of generating CREAM tokens (cf. Section 3.2) can be more concise. Meanwhile, more motivation about each procedure step could also help to understand the design.

**Strength And Weaknesses:**

## Strengths
+ The direction of designing a better tokenization method for Transformer-based SGG model is interesting.

## Weaknesses
+ The contributions are incremental. Based on my understanding, the only technique contribution is the new token generation strategy for the Transformer structure.

+ The motivations or logicality of the CREAM tokens designs are not clear. Specifically, in Sec.3.2, although the paper detailedly introduces the procedures to generate the CREAM tokens (cf. Figure 2), it is still unclear the advantages of this design. In another word, why this design "explicitly fuse each embedding token with known dependencies, which acts as a strong inductive bias for SGG".

+ The comparison between the state-of-the-art methods (Table 1 and Table 2) is unfair and incomplete. Firstly, it is unfair to directly compare these SGG methods with different privileged information. Secondly, compared with debiased methods separately in Table 2 is unreasonable. In contrast, it would be more complete to compare all SGG methods (in both Table 1 and Table 2) with both mRecall@K and Recall@K metrics. Thirdly, many strong baselines are missing.

+ Based on Figure 1, the difference between primal fusion and early fusion is not obviously. For example, in the early fusion case, start from the second layer, all the tokens features are fused from both modalities and subject-object features.

**Summary Of The Paper:**

This paper focuses on Scene Graph Generation (SGG), a task that aims to predict the relationships between objects detected in a scene. In this paper, they propose "primal fusion" which tries to inject entity relation information (between the features of the subject and object entity) and modality dependencies (between RGB and the corresponding depth modality) into each embedding token of the Transformer. The authors argue that this Cross-RElAtion-Modality (CREAM) token acts as a strong inductive bias for the SGG framework. Results on two benchmarks (Visual Genome and Open Image) are reported to show the effectiveness of the proposed method.

**Summary Of The Review:**

Generally speaking, the basic idea of designing better tokens for Transformer-based SGG models is interesting. However, the whole motivation or philosophy of each step is not clear, ie, the writing quality needs to be further improved. Meanwhile, the comparison between the state-of-the-art methods is not convincing, which has missed lots of stronger baselines.

---

### Official Review · Reviewer_adsx · 2022-10-26

**Confidence:** 4
**Correctness:** 3
**Technical Novelty And Significance:** 3
**Empirical Novelty And Significance:** 3
**Recommendation:** 5

**Clarity, Quality, Novelty And Reproducibility:**

The paper and the explanation is overall clear. However, the discussion on the idea of using Self-attention for Scene Graph Generation modeling is not addressed enough, with missing related work(s). Unless the proposed method and the core idea is not clearly differentiated to the previous method, the novelty of the paper is also not clear. Also, the SGDET score of the method is slightly worse than the existing methods, which makes it not clear what the advantage of not using the image context is. Is it very critical to avoid using image context, while it can provide a very important information for the overall SGG detection performance?

**Strength And Weaknesses:**

Strength
+ An extensive ablation study and analysis is provided
+ it achieves the state-of-the-art performance on the PREDCLS task.
+ The methods does not rely on the image context, while performing in a reasonable level.

Weakness
- The use of self attention to capture the relationship between subject-object pairs, which is argued as the core idea of this paper, has been proposed in the previous scene graph generation methods. One of the references is “LinkNet: Relational Embedding for Scene Graph, NeurIPS 2018”. There is no discussion and citation in the paper regarding these previous SGG methods based on the self attention mechanism.
- The proposed method shows worse SGDET score compared to the KERN and GB-Net methods. SGDET is the ultimate form of the Scene Graph Generation, which I believe is the most important metric among the PREDCLS, SGCLS, and SGDET.


**Summary Of The Paper:**

The paper tackles the Scene Graph Generation (SGG) problem, with a focus on resolving biased and under-represented relationship classes. The proposed method is built upon a multi-modality SGG model where a Transformer is used to model the relationship information and modality dependencies. The proposed method is tested on the Visual Genome and Open Images datasets and improve over the state-of-the-art baselines.

**Summary Of The Review:**

As addressed above, the novelty of the paper is not clear unless the discussion with the related work is explained. Also, the motivation behind not using the image context, and using multi-modality instead is not motivated well enough.

---

### Author Response · Authors · 2022-11-17
**Participation in discussion**

Dear reviewers,

We hope to have answered your concerns in our individual responses to the provided reviews. We would be happy to answer any further queries you might have before the end of the discussion period. Do let us know if you found our response satisfactory or/and wish to take forward the discussion.

Regards,

The Authors

---

### Decision · Program_Chairs · 2023-01-20

**Decision:**

Reject

**Justification For Why Not Higher Score:**

Reviewers raised some valid concerns on paper writing and experiments, and they have consistent negative ratings. Reviewers found that the responses could not fully address their concerns.

**Justification For Why Not Lower Score:**

N/A

**Metareview: Summary, Strengths And Weaknesses:**

This paper aims to exploit multi-modality for transformer based scene graph generation by using cross-relation-modality (CREAM) tokens. The goal is to encourage the transformers to learn enriched feature representations. Empirical results on benchmark datasets are reported and discussed.

Overall, the paper is well organized, and designing a better tokenization method for Transformer-based SGG model is a promising direction. The authors have conducted extensive evaluations and ablation studies.

Meanwhile, there are some limitations in the current version, such as the limited novelty, unclear motivation, and the claims in experiments. The authors have provided detailed responses, which addressed some of the concerns from reviewers. However, during the discussion phase, reviewers pointed out some remaining issues such as the novelty of the proposed idea and the claim on SOTA performance. Hopefully these issues could be addressed in the next version of this work.

**Summary Of Ac-Reviewer Meeting:**

N/A